# Rethinking MUSHRA: Addressing Modern Challenges in Text-to-Speech Evaluation

## Abstract

Despite rapid advancements in TTS models, a consistent and robust human evaluation framework is still lacking. For example, MOS tests fail to differentiate between similar models, and CMOS's pairwise comparisons are time-intensive. The MUSHRA test is a promising alternative for evaluating multiple TTS systems simultaneously, but in this work we show that its reliance on matching human reference speech unduly penalises the scores of modern TTS systems that can exceed human speech quality. More specifically, we conduct a comprehensive assessment of the MUSHRA test, focusing on its sensitivity to factors such as rater variability, listener fatigue, and reference bias. Based on our extensive evaluation involving 471 human listeners across Hindi and Tamil we identify two primary shortcomings: (i) *reference-matching bias*, where raters are unduly influenced by the human reference, and (ii) *judgement ambiguity*, arising from a lack of clear fine-grained guidelines. To address these issues, we propose two refined variants of the MUSHRA test. The first variant enables fairer ratings for synthesized samples that surpass human reference quality. The second variant reduces ambiguity, as indicated by the relatively lower variance across raters. By combining these approaches, we achieve both more reliable and more fine-grained assessments. We also release MANGO, a massive dataset of 47,100 human ratings, the first-of-its-kind collection for Indian languages, aiding in analyzing human preferences and developing automatic metrics for evaluating TTS systems.

## 1 Introduction

Human evaluation is widely regarded as the gold standard for Text-To-Speech (TTS) assessment; however, it lacks standardization. This issue is more realized with the rapid advancements in TTS synthesis, where numerous models claim superiority over prior systems or human speech Li et al. (2023); Wang et al. (2023); Tan et al. (2024). Deciphering the true extent of improvement from one model to the next is highly challenging due to inconsistent and often inadequately described subjective evaluation methodologies across studies.

The above problem is well studied for the Mean Opinion Scores (MOS) test Wester et al. (2015); Finkelstein et al. (2023); Kirkland et al. (2023) which has received much constructive criticism over the past few years. Specifically, in a MOS test, listeners assess each system independently, which can result in an inability to accurately capture the subtle relative differences between similar systems. This poses a significant challenge in modern TTS evaluation where systems that perform equally well need to be compared against each other. To address these issues some of the recent works rely on CMOS tests Loizou (2011). However, this test is costly and time-consuming as it involves $\binom{N}{2}$ comparisons between all pairs of $N$ systems.

The MUSHRA test has gained popularity in addressing these issues. This test scales better by enabling a parallel comparison of the $N$ systems, and addresses the limitations of MOS tests that only allow isolated evaluation. However, we show that even the MUSHRA test is not devoid of issues. To begin with, we note that the MUSHRA test was conventionally designed to assess intermediate-quality audio systems ITU-R (2015). However, state-of-the-art TTS systems Ju et al. (2024) are not of intermediate quality and instead generate audios having quality on par or even better than human recordings. To align with these modern developments, several works adopt variants of MUSHRA

Merritt et al. (2022); Li et al. (2023); Shen et al. (2024), which differ in implementation but the validity of these modified tests is unknown.

Given this situation, we critically assess the reliability, sensitivity, and validity of the MUSHRA tests by asking a series of research questions, such as: Is MUSHRA a reliable test, consistently yielding results comparable to other widely adopted subjective tests such as CMOS? Is the mean statistic reported in MUSHRA reliable, or is there significant variance across listeners and utterances? How sensitive is MUSHRA to implementation details? Particularly, how many listeners and utterances are required to yield statistically significant results? Is the conventional MUSHRA reject rule appropriate when TTS outputs sometimes outperform ground-truths? How does the choice of anchor affect MUSHRA scores, and what is the optimal anchor? While some of these questions have been studied for MOS Wester et al. (2015), a comprehensive assessment of MUSHRA remains lacking.

With the goal of seeking answers to the above questions, we collected 47,100 human ratings by conducting the MUSHRA test involving 3 systems across two languages, viz., Tamil and Hindi. Our in-depth analysis based on these ratings, reveals two primary shortcomings: (i) reference-matching bias and (ii) judgement ambiguity. To mitigate these issues, we propose two refined variants of the MUSHRA test. The first variant does not explicitly identify the human reference to the rater. Doing so, prevents unfair penalties for well-synthesized samples that differ from the human reference, such as those with natural prosody that do not match the reference's prosody. In the second variant, raters are provided scoresheets to systematically calculate MUSHRA scores, by explicitly marking pronunciation mistakes, unnatural pauses, digital artifacts, word skips, liveliness, voice quality, rhythm, etc. Using the scores for these fine-grained criteria, they arrive at the final MUSHRA score. Our studies show that both these variants lead to a more reliable evaluation with the second variant also allowing for fine-grained fault isolation during evaluation. We then show that a combination of these two approaches that leverages their individual strengths ensures both consistency and granularity. It allows modern TTS systems to be evaluated without being unfairly penalized for surpassing the reference in naturalness or prosody. The detailed scoring for pronunciation, prosody, and other factors provides actionable insights,and helps practitioners understand precisely where their systems excel and where improvements are needed. This combination creates a more balanced and sensitive evaluation framework, offering a clearer and more reliable assessment of TTS system performance.

In summary, our main contributions are:

1. A comprehensive assessment of the reliability, sensitivity, and validity of the MUSHRA test implementation in evaluating modern high-quality TTS systems.

2. Identification of two primary shortcomings of the MUSHRA test: (i) Reference-matching bias and (ii) Judgement Ambiguity.

3. Proposal of two variants of MUSHRA aimed at addressing these shortcomings.

4. Large-scale empirical validation of proposed variants resulting in MANGO, a dataset of 47,100 ratings from 471 listeners across Hindi and Tamil, examining three TTS systems.

## 2 RELATED WORK

**Critiques of TTS Evaluation.** Prior works mainly focused on a critique of MOS tests. Wester et al. (2015) analyze results from the Blizzard Challenge 2013 and highlight that an adequate number of listeners and utterances are needed to accurately identify significant differences. Clark et al. (2019) find that MOS tests are context-sensitive and yield different results when evaluating sentences in isolation as opposed to rating whole paragraphs. MOS tests are also known to show high variance in ratings Finkelstein et al. (2023), subject to how raters are chosen. Kirkland et al. (2023) realize the importance of reporting scale labels, increments, and instructions, and show how these variables can affect scores. A recent study Cooper & Yamagishi (2023) highlights the presence of range-equalizing bias in MOS tests. Chiang et al. (2023) analyze over 80 papers, noting insufficient description of evaluation details and its impact on evaluation outcomes. Similarly, Le Maguer et al. (2024) highlight the need for better evaluation protocols.

**Emergence of Modern Tests.** Several variants of MUSHRA have been employed to overcome known shortcomings. To evaluate the robustness of TTS trained on imperfect transcripts, Fong et al. (2019), adopt the MUSHRA test without an anchor and also provide text transcripts dur-

ing evaluation. Taylor & Richmond (2020) measure impact of morphology using a hidden natural reference, and utterances containing out-of-vocabulary words. Aggarwal et al. (2020) extend the MUSHRA test to also measure emotional strength of the synthesised speech. Merritt et al. (2022) adopt MUSHRA for evaluating speaker and accent similarity, by including both an upper-anchor and lower-anchor along with hidden reference. Li et al. (2023) adopt a variant of the MOS test, similar to MUSHRA, for testing naturalness and speaker similarity.

**Learnings from Human Evaluations in NLP** Freitag et al. (2021) highlighted the need for comprehensive, standardized evaluation frameworks like Multidimensional Quality Metrics for MT, which is crucial for TTS too. Ethayarajh & Jurafsky (2022) show that the average of Likert ratings (as followed in MOS tests in TTS) can be a biased estimate potentially leading to misleading rankings. Amidei et al. (2019) discuss how insufficient descriptions can make it difficult to interpret evaluation results. Howcroft & Rieser (2021) emphasize that current evaluations are inadequate for detecting subtle distinctions between systems; a problem we find recurring in TTS evaluations. Direct assessments have been popular in WMT evaluations Barrault et al. (2020); Akhbardeh et al. (2021), however Knowles (2021) highlight several of its issues, a caution that carries over to human evaluations for TTS. They also advocate evaluating multiple systems on the same subset of documents, a practice we mirror in this work using audio samples instead.

## 3 MANGO: A CORPUS OF HUMAN RATINGS FOR SPEECH

We introduce a new dataset, MANGO: MUSHRA Assessment corpus using Native listeners and Guidelines to understand human Opinions at scale. It is a first-of-its-kind collection for any Indian language, comprising 47,100 human ratings of TTS systems and ground-truth human speech in both Hindi and Tamil. Given the shortcomings of Mean Opinion Score (MOS) and Comparative Mean Opinion Score (CMOS) tests, our goal is to critically examine a promising alternative—the MUSHRA test—by conducting a large-scale evaluation involving multiple raters, systems, and languages. To do so, we adopt the standard MUSHRA test ITU-R (2015). Raters evaluate multiple stimuli on each page, including an explicit (mentioned) that serves as a benchmark for high-quality speech, along with an anchor and implicit (hidden) reference to calibrate judgments. Each stimulus is rated on a continuous scale from 0 to 100, which is also discretized: 100-80 (Excellent), 80-60 (Good), 60-40 (Fair), 40-20 (Poor), and 20-0 (Bad). We describe our evaluation setup below, and provide the detailed instructions provided to participants in Appendix A.1.

### 3.1 ONLINE ANNOTATION PLATFORM

We enhance the webMUSHRA Schoeffler et al. (2018) platform to address its key limitations. Specifically, we modify a fork Pauwels et al. (2021) and introduce session management to enable saving test progress and thereby allowing for breaks for listeners. We integrate consent forms, and controls, such as ensuring raters listen to all audio samples in their entirety, to ensure more reliable ratings. We also integrate an event-tracking system to analyze time spent per page.

### 3.2 SYNTHESIZING SPEECH SAMPLES FOR ANNOTATION

To generate samples for TTS evaluation, we train TTS systems on the Hindi and Tamil subsets of the IndicTTS database Baby et al. (2016). Each language consists of recordings from a female and male speaker (Hindi: 20.17 hours; Tamil: 20.59 hours). We train FastSpeech2 (FS2) Ren et al. (2021) with HiFiGAN v1 Kong et al. (2020), VITS Kim et al. (2021) from scratch on the train-test splits using hyper-parameters suggested in a recent study Kumar et al. (2023). We finetune StyleTTS2 (ST2) Li et al. (2023) from the LibriTTS checkpoint and XTTSv2 CoquiAI (2023) from the multilingual checkpoint with the hyperparameters from their original implementations on the same splits.

### 3.3 ANNOTATION PROCESS AND DATASET STATISTICS

To ensure reliable evaluation, we recruited native speakers of the target languages through reputable recruitment agencies. These agencies played a vital role in guaranteeing participant demographics aligned with the target language of each test. Please refer to Section A.7 for details on recruitment, consent and compensation. Once recruited, the annotators underwent a comprehensive training pro-

Table 1: Dataset statistics of MANGO.

| Language | # Ratings | Gender | | Age | | | | | # Participants in MUSHRA Variants | | | |
|----------|-----------|--------|------|-------|-------|-------|-------|-----|----------|-----|----|--------|
| | | Female | Male | 18-25 | 25-30 | 30-35 | 35-40 | 40+ | Original | NMR | DG | DG-NMR |
| Hindi | 24,400 | 64 | 161 | 140 | 52 | 17 | 11 | 5 | 113 | 102 | 14 | 15 |
| Tamil | 22,700 | 146 | 79 | 78 | 73 | 36 | 26 | 12 | 100 | 97 | 15 | 15 |

Table 2: MUSHRA scores for Hindi and Tamil using Anchor-X and Anchor-Y, respectively, as anchors (ANC). $\mu$ represents the mean, $\sigma$ represents the standard deviation, and the $95\%$ confidence intervals (CI) are provided.

| System | Hindi | | | Tamil | | |
|--------|-------|-------|------|-------|-------|------|
| | $\mu$ | $\sigma$ | CI | $\mu$ | $\sigma$ | CI |
| FS2 | 64.17 | 22.89 | 0.42 | 64.98 | 19.23 | 0.38 |
| ST2 | 66.74 | 21.65 | 0.40 | 71.38 | 18.31 | 0.33 |
| VITS | 67.65 | 20.58 | 0.38 | 65.66 | 18.91 | 0.37 |
| ANC | 70.81 | 20.92 | 0.39 | 20.08 | 16.69 | 0.38 |
| REF | 84.18 | 15.49 | 0.29 | 85.22 | 15.98 | 0.31 |

Table 3: Mean Comparitive-Mean-Opinion-Scores (CMOS) with 95% confidence intervals for Hindi & Tamil.

| System | Hindi | Tamil |
|--------|-------|-------|
| REF | - | - |
| ST2 | -0.11 ± 0.08 | 0.24 ± 0.09 |
| VITS | -0.10 ± 0.07 | -0.57 ± 0.09 |
| FS2 | -0.66 ± 0.08 | -0.60 ± 0.09 |

cess comprising multiple sessions aimed at familiarizing them with the evaluation platform, test interface, and evaluation criteria. Guidelines were clearly explained, and any doubts were addressed to ensure that all participants had a uniform understanding of the evaluation process. Additionally, a structured review process was implemented, wherein participants initially rated five pilot samples. This phase allowed them to seek clarifications, provide feedback, and ensure their understanding of the evaluation criteria before proceeding to rate the 100 test samples. This approach not only improved their confidence but also helped to standardize the assessment process across all evaluators.

With the above process we collected 47,100 human ratings for TTS systems. Table 1 shows the demographic distribution, the number of participants and the overall number of ratings across all MUSHRA tests and our proposed variants (MUSHRA-NMR, MUSHRA-DG, MUSHRA-DG-NMR) which are described in Section 5.

## 4 KEY INSIGHTS ON MUSHRA

In this section, we address the research questions outlined in Section 1 and identify key challenges based on ratings collected in the MANGO dataset.

### 4.1 IS MUSHRA A RELIABLE TEST?

In Table 2, we present the results of the MUSHRA test among 3 systems and find VITS and ST2 score highest in Hindi and Tamil respectively. Surprisingly, all systems attain scores in the "Good" bin with MUSHRA scores between 60 and 80, while the reference surpasses all systems with scores in the "Excellent" bin. Given that state-of-the-art TTS systems are able to reach quality on par with references, one would expect a much smaller gap between the reference and systems. To confirm this, we conduct the more reliable but expensive CMOS test with 15 listeners in each language. In this test, we ask the rater to compare a given system, such as VITS, with a reference audio sample. The rater evaluates both the reference and the output from a system being tested without prior knowledge of which audio sample corresponds to which system, ensuring an unbiased comparison. The raters assign a single score ranging from -3 to +3 in increments of 0.5. A score of -3 indicates that System A is much worse than System B . A score of +3 indicates that System A is much better than System B. A score of 0 means that both systems are equal in quality. As seen from the scores in Table 3, CMOS indicates that the outputs synthesised by VITS and ST2 are very close in quality to the reference in Hindi and Tamil respectively, while MUSHRA scores do not reflect this at all. We hypothesize that listeners in the MUSHRA test are subject to various biases, one of which we term the *reference-matching bias*. This bias may lead to situations where systems that perform compara-

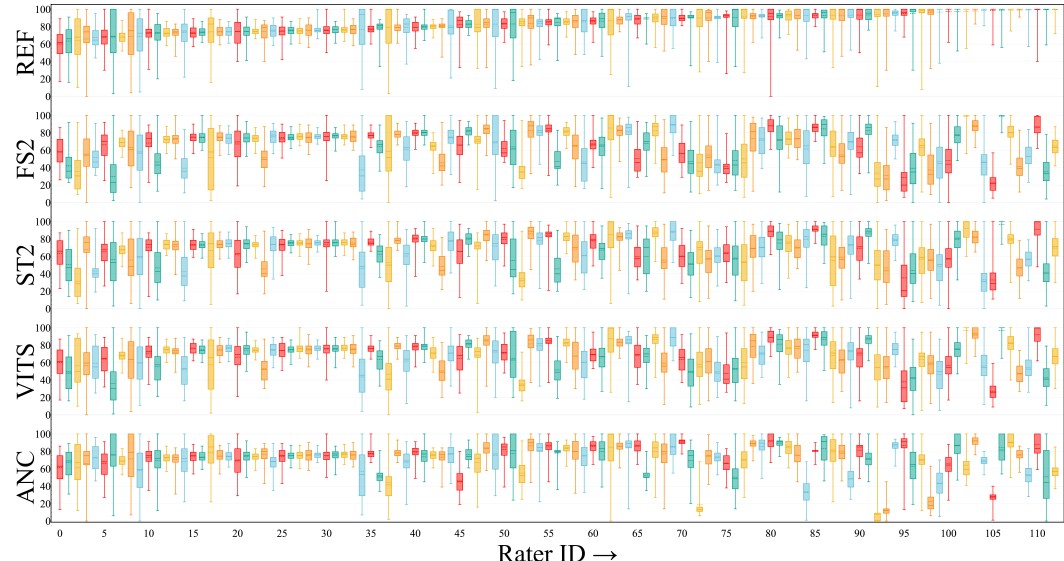

Figure 1: Visualization of the MUSHRA score distributions per rater across three systems— FS2, ST2, and VITS, along with the reference (REF) and anchor (ANC) for Hindi. Each boxplot represents ratings (0-100) across all test utterances for a system by one rater. The substantial heights of some boxplots indicate significant variance in the scores of that rater. The variation in the means of the boxplot across raters suggests a high level of inter-rater variance. Raters are sorted in ascending order of their mean scores for the reference.

bly to or better than the reference are rated less favorably, as listeners tend to focus on aligning their ratings with the reference outputs while evaluating the systems. While this may have been acceptable when TTS systems lagged behind human speech quality, it is undesirable in the current scenario where modern TTS systems often exceed the reference in aspects like naturalness and prosody. This suggests that the MUSHRA test, in its conventional form, may no longer be sufficient for evaluating state-of-the-art TTS systems. Instead, alternative methodologies, such as the variants we propose in Section 5, may help ensure more fair and accurate assessments.

### 4.2 How reliable is the mean statistic in MUSHRA scores?

As mentioned earlier, each rater rates 100 utterances. In Figure 1, we use box-plots to visualize the distribution of MUSHRA scores (y-axis) for each rater (x-axis) across these utterances for each system, including the reference and anchor. While we acknowledge that the figure may appear overwhelming, we believe it is crucial for conveying the comprehensive view across both raters and utterances. We make two important observations from the figure. First, the individual box-plots have a high variance indicating that the same rater rates the system very differently across utterances. Second, looking at the means of the box-plots across different raters, we observe that there is a high variance in the means, indicating ambiguity in the perception of the MUSHRA labels across raters. We refer to this phenomenon as *judgement ambiguity*. This highlights the shortcomings of reporting mean statistics for MUSHRA scores, even when reported with confidence intervals (CI).

To delve deeper into *judgment ambiguity*, we examine variations between two systems. We consider an utterance where the mean scores for the samples generated by VITS and ST2 are nearly identical, but the variance across raters for each system is high. This high variance indicates significant ambiguity. Upon listening to many such utterances and speaking to many raters, we hypothesize that the ambiguity likely stems from different raters focusing on different aspects of the generated samples. For instance, some raters may prioritize prosody, others voice quality, and yet others the presence of digital artifacts. We hypothesize that asking raters to highlight these subtle differences across multiple dimensions while assigning a single score can lead to ambiguity in determining how much to penalize or reward a system's score. Hence, clear guidelines which take into account a fine-grained evaluation across different aspects would help (as proposed later in Section 5).

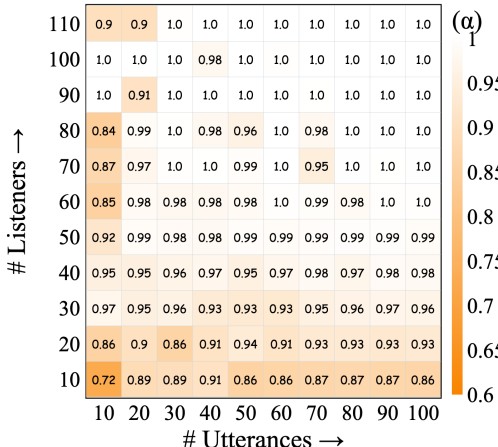

Figure 2: Rank correlation of mean scores obtained using subsets of listeners and utterances and mean scores obtained using all listeners and utterances in Hindi.

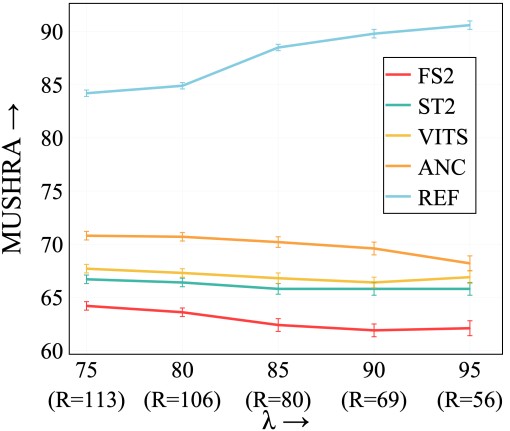

Figure 3: MUSHRA Scores in Hindi show score-variance but rank-invariance across systems when raters who rate Reference $\leq \lambda$ for more than 15% of utterances are rejected. R is the number of raters retained.

### 4.3 HOW SENSITIVE IS MUSHRA TO NUMBER OF LISTENERS AND UTTERANCES?

We use the procedure outlined in Wester et al. (2015) to study the effect of number of listeners and utterances on MUSHRA scores. Specifically, we are interested in knowing if a smaller number of listeners and utterances would result in the same rankings of systems as obtained using the full set of listeners and utterances. To achieve this, we randomly sample a smaller subset of utterances and listeners and compute the mean system scores. We then calculate the Spearman rank correlation with the mean system scores obtained using all utterances and listeners. We repeat this process 1000 times, and compute the average over these large number of trials.

Figure 2 illustrates the average correlation of MUSHRA ratings in Hindi between a subset of listeners and utterances compared to the fully-scaled test (involving all listeners and utterances). Firstly, it is evident that using a minimum of 20 listeners is crucial to achieve correlations above 90%. Secondly, when employing a smaller number of listeners and utterances (e.g., fewer than 40 in both cases), increasing the number of listeners proves to be more beneficial than increasing the number of utterances. Using more than 30 listeners and 30 utterances invariably yields correlations above 95%. We notice similar trends for Tamil, but with higher correlations achieved with lesser number of listeners and utterances (Appendix 8). These results highlight the sensitivity of MUSHRA evaluations to the number of listeners and utterances, emphasizing the importance of careful selection and scaling of these factors to ensure reliable and meaningful evaluations.

### 4.4 WHAT IS THE IMPACT OF REJECTING RATERS PER STANDARD MUSHRA PROTOCOL?

Traditionally, MUSHRA employs a rater rejection criterion, wherein raters scoring the hidden reference (HR) below a threshold ($\lambda$) more than 15% of the time are rejected. This rejection rule stems from the inherent assumption that the HR is the gold standard, which is not true in modern TTS settings where TTS systems Li et al. (2023) claim to achieve performance surpassing the reference. In such cases, a rater consistently scoring the HR lower might not necessarily reflect unreliability, but rather a nuanced perception of the reference's limitations compared to the evaluated systems. This is reinforced by observations from Table 3 where raters clearly prefer ST2 over reference for Tamil with a mean CMOS score of 0.24. This observation is further supported by the MUSHRA scores shown in Figure 3. The table shows that while rejecting raters based on the conventional MUSHRA criterion does not affect system rankings, it does notably shift scores. Specifically, system scores decrease with increasing $\lambda$, while reference scores increase. This trend hints at the *reference-matching bias* wherein raters who give the HR high scores might be unconsciously matching system samples to the mentioned reference, rather than rating based on their absolute perception of quality.

### 4.5 How does Anchor affect scores?

In MUSHRA tests, the anchor serves the purpose of setting the expectation of what a Fair sample sounds like. Typically, the anchor is created by minimally degrading the ground-truth by first down-sampling it to 3.5 kHz and then upsampling it to 24 kHz. We refer to such an anchor as Anchor-X. In Table 2, the mean scores for Anchor-X in Hindi indicate that this anchor performs significantly better than all other systems, attaining a high score of 70.81. We believe these scores are explained by the resampling strategy used to create the Anchor-X, which introduces some artifacts in the audio but retains similar naturalness to the reference, especially in terms of prosody. Once again, this intuition indicates the tendency of raters to rate systems that match the reference with higher scores (reference-matching bias). We conclude that using this anchor may not be ideal, as it can lead to potentially "Excellent" TTS systems being unfairly rated as "Good". Essentially, the raters may perceive that if one of systems (anchor, in this case) sounds very similar to the reference, there is little justification for rating other systems highly.

Next, we study the use of an alternative anchor (Anchor-Y) that we know would likely fall in the "Poor" or "Fair" category, given its construction process. Specifically, we construct Anchor-Y by degrading ST2 outputs by averaging the pitch, reducing the number of diffusion steps, slowing the audio by 1.2 times, and inducing mispronunciation via the input text, along with word skips and word repeats in 20% of the samples. To obtain average voice quality, we set the number of diffusion steps to 3 with $\alpha = \beta = 0.8$. As expected, from the Tamil MUSHRA scores in Table 2, this anchor does indeed score poorly with a mean of 20.08. Interestingly, we see that despite a very low-quality anchor, other systems are not rated very highly, and there is still a huge disparity between ST2 (71.38) and the Reference (85.22). Given that high-quality anchors unfairly bias raters against other systems, while low-quality anchors seem to have no effect on the ratings for other systems, we believe there is merit in conducting MUSHRA evaluations without anchors Lajszczak et al. (2024), which also saves costs by reducing human effort.

### 4.6 Does adding more systems affect scores?

To better understand cognitive overload in the MUSHRA test, we scale up the number of systems to be rated by introducing one new competitive system - XTTS, and repeating an existing system - VITS (VITS-R) in the original Hindi MUSHRA test. We call this MUSHRA-Extended. The results show that raters were highly consistent, with VITS and VITS-R receiving nearly identical scores (68.99 and 68.47, respectively), despite the randomized order. Introducing XTTS, which outperformed other systems with a score of 73.65, did not disrupt the relative ranking of the remaining systems, which remained consistent with the original MUSHRA test. Thus, there does not seem to be significant cognitive overload, as we still observe consistent results. Note that we study cognitive load using $n = 7$ systems, but it remains to be seen how large $n$ can be before cognitive overload starts impacting the scores. For detailed scores, please refer to Table 5 in appendix.

## 5 Rethinking MUSHRA

We summarize two issues identified in Section 4. First, *reference-matching bias* that arises when listeners rate systems that perform at or above the level of the reference lower than deserved due to their efforts to align system outputs with the reference during evaluation. Second, *judgement ambiguity* that arises when listeners rate a system on a single scale using broadly defined metrics like "naturalness", leaving room for subjective interpretation of sub-criteria such as "prosody", "voice quality", "liveliness", etc. leading to high variability in ratings. In response to this, we propose two refined variants of the MUSHRA test to address the identified challenges, as described below.

**MUSHRA-NMR.** The first variant, MUSHRA-NMR (MUSHRA with No Mentioned Reference), aims to mitigate the reference-matching bias observed in our analysis. MUSHRA-NMR follows all other standard protocols of the MUSHRA ITU-R (2015) test, except for the omission of the explicitly mentioned ground-truth reference that is presented to the listener. In the absence of this explicilty mentioned reference, the listener will be able to independently assess the quality of the TTS systems without trying to match them against the reference.

**MUSHRA-DG.** The second variant, MUSHRA-DG (MUSHRA With Detailed Guidelines), introduces comprehensive guidelines to reduce the ambiguity in rating samples for naturalness. In this

test, we present raters with scoresheets and a formula to arrive at MUSHRA scores systematically. Each rater was asked to mark the number of (i) mild pronunciation mistakes, (ii) severe pronunciation mistakes, (iii) unnatural pauses, speedups, or slowdowns, (iv) digital artifacts, (v) sudden energy fluctuations, and (vi) word skips. Further, raters were also asked to rate more perceptual measures such as (i) liveliness, (ii) voice quality, and (iii) rhythm on a continuous scale from 0-100. The detailed guidelines provided to raters to assess across each of these dimensions can be found in Appendix A.3. We analytically derive a MUSHRA naturalness score to raters using an intuitive formula with weights (provided in Appendix A.3) for different dimensions listed above. These weights can be tweaked depending on the specific use-case. For example, in a TTS application designed for audiobooks, where fluidity and expressiveness are crucial for user engagement, we might assign higher weights to liveliness and rhythm.

We understand that devising a scoring formula involves some subjectivity. To address this, we encouraged raters to review their evaluations and adjust their fine-grained scores if they feel the overall scores from the formula did not accurately reflect the differences they perceived between system pairs. This way, the final ratings better represented the raters' true opinions about each system's quality and reduce any shortcomings that could have stemmed from the formula. More interestingly, we notice that the scores derived from the MUSHRA-DG test preserve the rankings obtained from the gold-standard Comparative Mean Opinion Score (CMOS) tests (Table 3), thus reinforcing the validity of our evaluations. Additionally, the variance in MUSHRA scores calculated using the formula is significantly lower, indicating reduced ambiguity in ratings and providing a clearer distinction between different systems' performances.

## 6 RESULTS

We present human evaluation results of our proposed MUSHRA variants from the MANGO dataset.

### 6.1 EVALUATIONS USING MUSHRA-NMR

**Does our proposed variant help mitigate the reference-matching bias?**

In Table 4, we present the results of the MUSHRA-NMR test. We find the results to be rank-consistent with the scaled-up MUSHRA tests (Table 2). In the case of Tamil, we observe that the best performing system (ST2) is now scored much closer to the reference, clearly suggesting that the reference-matching bias has been mitigated. We observed that the score assigned to the reference itself decreased, indicating that the raters were strict. In the case of Hindi, the gap between the best performing system and the reference has again decreased but is not as small as in the case of Tamil.

We want to re-emphasize that this expectation of a reduced gap between the system and reference scores is well-founded. Feedback from TTS practitioners, including some of the authors who are native speakers, revealed that while some of the systems performed impressively in practice, the original MUSHRA scores did not seem to fully reflect their quality. This shortcoming is also clearly seen by the significant score differences between the reference and the other systems in the original MUSHRA test, whereas the CMOS scores in Table 2 indicate a closer alignment of a system's performance with the human ground-truth reference. Collectively, our findings above reinforce the merits of our proposed MUSHRA-NMR variant, which offers more reliable relative assessments compared to the MUSHRA test while retaining the advantages MUSHRA has over the CMOS test.

**How sensitive is MUSHRA-NMR to number of listeners and utterances?**

Subjective evaluations are often resource-intensive, making it desirable to minimize the number of listeners and utterances without compromising assessment quality. To explore this, we present the correlation between the scores derived from the MUSHRA variants using a subset of listeners and the scores obtained from the complete listener set, as shown in Figure 4. We also do a similar comparison across utterances. Our findings reveal that MUSHRA-NMR achieves a Spearman rank correlation exceeding 95% with the fully scaled-up MUSHRA test using just 20 utterances or 40 listeners. This indicates that significant reductions in both parameters are possible while maintaining reliability. Interestingly, our analysis indicates that enhancing the number of listeners has a greater impact on the accuracy of assessments compared to simply increasing the number of utterances.

Table 4: Comparison of MUSHRA scores and proposed variants for Hindi and Tamil languages.

| Language | System | MUSHRA-NMR | | | MUSHRA-DG | | | MUSHRA-DG-NMR | | |
|---|---|---|---|---|---|---|---|---|---|---|
| | | $\mu$ | $\sigma$ | 95% CI | $\mu$ | $\sigma$ | 95% CI | $\mu$ | $\sigma$ | 95% CI |
| Hindi | FS2 | 61.99 | 23.86 | 0.46 | 73.02 | 12.05 | 0.63 | 85.76 | 8.38 | 0.42 |
| | ST2 | 68.09 | 22.01 | 0.43 | 73.03 | 11.87 | 0.62 | **89.55** | 8.29 | 0.35 |
| | VITS | **68.75** | 21.04 | 0.41 | **75.08** | 10.93 | 0.57 | 89.33 | 7.11 | 0.42 |
| | Anchor-X | 71.83 | 19.97 | 0.39 | 77.67 | 14.60 | 0.76 | 89.73 | 6.96 | 0.38 |
| | Reference | 76.39 | 18.08 | 0.35 | 91.45 | 10.30 | 0.54 | 89.45 | 7.47 | 0.36 |
| Tamil | Anchor-Y | 21.94 | 16.74 | 0.38 | 45.63 | 10.14 | 0.35 | 56.01 | 7.46 | 0.34 |
| | FS2 | 66.77 | 19.12 | 0.35 | 82.32 | 6.93 | 0.39 | 78.01 | 6.79 | 0.32 |
| | VITS | 68.52 | 18.28 | 0.36 | 82.32 | 5.60 | 0.28 | 78.33 | 6.39 | 0.32 |
| | ST2 | **76.64** | 17.68 | 0.33 | **88.50** | 7.72 | 0.51 | **88.77** | 6.33 | 0.38 |
| | Reference | 78.69 | 17.26 | 0.34 | 93.18 | 7.34 | 0.37 | 95.47 | 10.62 | 0.54 |

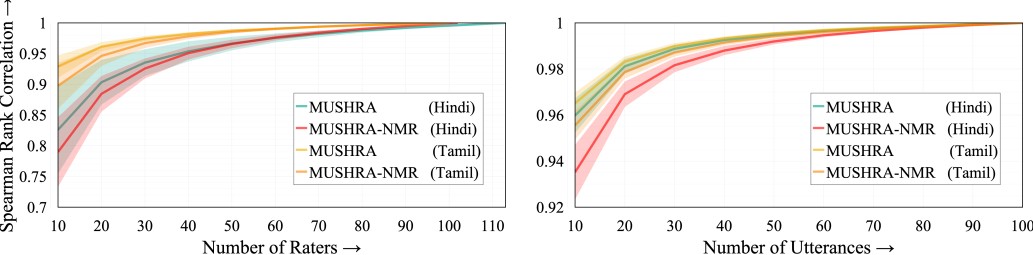

Figure 4: (Left) Correlation between scores from a subset of listeners and all listeners. (Right) Correlation between scores from a subset of utterances and all utterances.

## 6.2 EVALUATIONS USING MUSHRA-DG

**Does our proposed variant help mitigate judgement ambiguity while rating?** In Table 4, we show the effects of presenting detailed guidelines along with scoresheets to 14 participants to systematically arrive at MUSHRA scores. We find MUSHRA-DG scores to be rank-consistent with the CMOS tests , while systems scores are much higher and closer to the "Excellent" label, as expected. More importantly, the standard deviation of scores across all systems reduced by 41% in Hindi and 58% in Tamil when compared to the original MUSHRA, indicating that our proposed variant is able to reduce the ambiguity of rating naturalness on a single bar while preserving ranks.

**Fault Isolation.** We collate the scoresheets of participants to obtain more fine-grained insights on where each model underperforms. In Figure 5a, we report the error rates of instances where an attribute received a rating greater than 0 for the six objective attributes and in Figure 5b the

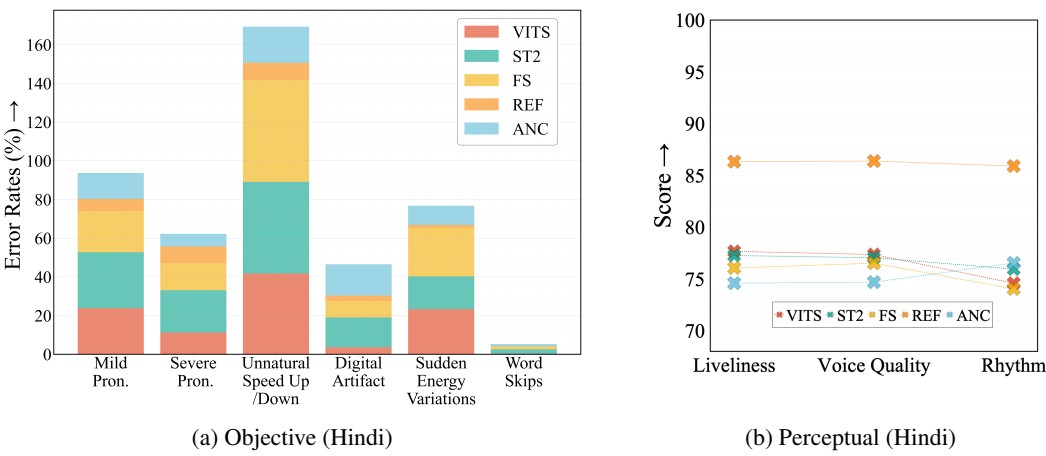

(a) Objective (Hindi)

(b) Perceptual (Hindi)

Figure 5: Visualization of the 6 objective and 3 perceptual dimensions of the MUSHRA-DG test.

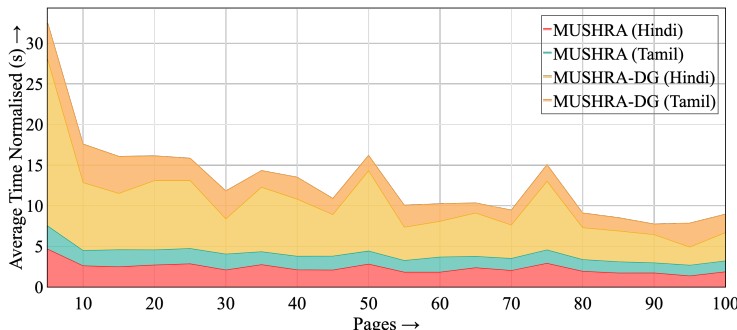

Figure 6: MUSHRA-DG exhibits higher average time (normalized by audio durations) across pages compared to MUSHRA.

absolute perceptual scores on a scale of 100 for the remaining attributes. The granular ratings reveal the true power of this test in identifying defects in TTS outputs, especially among systems that achieved similar mean scores in the original MUSHRA. Specifically, we observe that for Hindi, a deterministic system like FS2 performs well in terms of pronunciation but suffers in prosody and word-skipping. Conversely, the close difference between VITS and ST2 is better explained by noting that VITS nearly outperforms in all dimensions, except that VITS exhibits nearly twice as many sudden energy fluctuations as ST2 and performs slightly worse in terms of rhythm.

**Time Complexity of MUSHRA-DG.** We hypothesize that the additional detail of evaluating each audio sample across multiple dimensions inevitably increases the time required for participants to complete the test. To verify this hypothesis, we visualize the average time taken across pages in Figure 6 and find that the MUSHRA-DG test indeed takes nearly twice as much time as the original MUSHRA test. However, we believe this extra time results in a much more comprehensive understanding of TTS system performance, making the trade-off worthwhile.

### 6.3 EVALUATIONS USING MUSHRA-DG-NMR

In Table 4, we present the results of our combined variant, wherein we provide detailed guidelines (DG) and remove the mentioned reference (NMR). We observe that the majority of system scores now align more closely with the reference ratings and predominantly fall within the "Excellent" category, as anticipated from the CMOS tests. Moreover, compared to the MUSHRA-NMR test, the variance in scores has significantly diminished, indicating a marked reduction in rating ambiguity.

As TTS practitioners and native speakers of the language, we would like to emphasize that, despite the relative rankings being preserved in nearly all variants of the test, the combined variant is more reliable because the scores now reflect the expected proximity of the systems to the reference (also established by the CMOS scores).

## 7 CONCLUSION

Our comprehensive study reveals significant shortcomings in the current use of the MUSHRA test for evaluating modern high-quality TTS systems. Through an extensive analysis involving 47,100 human ratings, we identified two primary issues: reference-matching bias and judgment ambiguity. To address these issues, we propose two refined variants of the MUSHRA test: MUSHRA-NMR, which omits explicit identification of the human reference, and MUSHRA-DG, which uses detailed guidelines to calculate MUSHRA scores systematically. Our findings indicate that both variants lead to more reliable evaluations, with MUSHRA-DG offering the additional benefit of fine-grained fault isolation during assessment. Through this work, we also release MANGO, a large human rating dataset, to further support research in this area.

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

# A  APPENDIX

## A.1  INSTRUCTIONS FOR MUSHRA

**Instructions to Participants for MUSHRA Evaluations of Text-to-Speech Systems**

Thank you for participating in this speech evaluation study to assess the quality of various Text-to-Speech (TTS) systems. Please follow the instructions given below carefully.

**Overview**

In this evaluation, you will listen to different audio samples produced by various TTS systems. Your task is to rate these samples based on specific criteria using the MUSHRA (Multiple Stimuli with Hidden Reference and Anchor) methodology.

**Evaluation Procedure**

1. Listening Setup:
   a. Please use good-quality headphones or speakers to ensure you can hear all the nuances in the audio samples.
   b. Find a quiet space to minimise distractions during the evaluation.
   c. Use a consistent playback device throughout the evaluation to maintain uniformity in listening conditions.

2. Rating Scale:
   a. You will use a scale from 0 to 100 to rate the quality of each audio sample.
   b. The ratings correspond to the following categories:
      - 100-80: **Excellent**
      - 80-60: **Good**
      - 60-40: **Fair**
      - 40-20: **Poor**
      - 20-0: **Bad**

3. Listening and Rating:
   a. General Procedure: For each rating page in the MUSHRA test -
      I. Listen to the mentioned reference carefully to understand high quality.
      II. Then, listen to each system output. You can listen to samples multiple times if needed.
      III. Ensure you listen to each audio sample in its entirety without interruptions.
      IV. After listening to each sample, rate the quality of each of them based on its naturalness and overall quality.
      V. Please keep in mind that you can adjust your ratings as you listen to different samples.
      VI. Please take regular breaks after every 30 minutes to avoid strain and fatigue.
   b. Evaluation Criteria: After listening to each sample, rate the quality based on its naturalness and overall quality. Consider factors such as:
      - Naturalness: How similar does the audio sample sound to human speech?
      - Intelligibility: Is the speech clear and easy to understand?
      - Prosody: Does the output have appropriate intonation, rhythm, and stress?
   c. Comparative Assessment: Compare each sample with the others on the same page. Ensure that your ratings reflect the true relative rankings of the systems based on your perception. Your evaluations should capture the differences in quality as accurately as possible.

   d. Finalising Your Ratings:
      - Once you have rated all samples for a page, you may move to the next page.
      - Ensure that you are satisfied with your ratings before submitting, as they will be recorded.

If you have any questions or need assistance during the evaluation, please feel free to ask.

Figure 7: Guidelines sent to participants taking the MUSHRA test. They were given a live demo of the rating page and walked through the guideline sheet.

## A.2  SENSITIVITY OF MUSHRA IN TAMIL

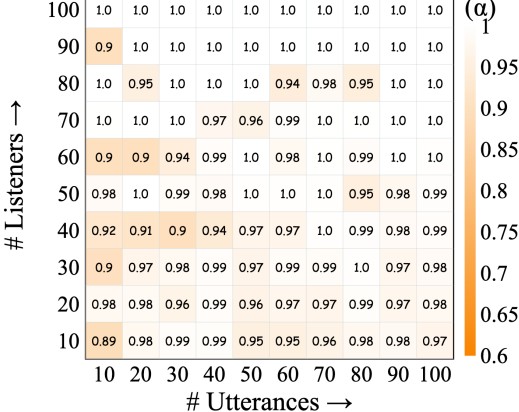

Figure 8: Spearman rank correlation of MUSHRA scores in Tamil using subsets of listeners and utterances vs. scores of all listeners and utterances.

### A.3 DETAILED GUIDELINES FOR MUSHRA-DG

We present the complete guidelines shown to raters in Figure 9. We derive a formula that takes into account several factors: mild pronunciation mistakes ($MP$), severe pronunciation mistakes ($SP$), unnatural speedup or slowdown ($US$), liveliness ($L$), voice quality ($VQ$), rhythm ($R$), digital artifacts ($DA$), sudden energy fluctuations ($SEF$), and word skips ($WS$). The MUSHRA score is calculated by averaging the perceptual measures and then penalizing for various mistakes and artifacts. Specifically, we penalize every word skip by deducting 25 points, every severe pronunciation mistake by deducting 10 points, and every mild pronunciation mistake by deducting 5 points. Likewise, all other non-perceptual measures are penalized by 5 points. The MUSHRA score ($s_M$) for a system is given by,

$$
\begin{aligned}
s_M = {} & \frac{L + VQ + R}{3} \\
& - \min(MP, 15) \times 5 \\
& - \min(7, SP) \times 10 \\
& - US \times 5 \\
& - DA \times 5 \\
& - WS \times 25 \\
& - SEF \times 5
\end{aligned}
$$

### A.4 MUSHRA-EXTENDED

Table 5: MUSHRA-Extended scores with 95% CI for Hindi.

| System | $\mu$ | $\sigma$ | CI |
|--------|-------|----------|------|
| FS2 | 63.12 | 21.3 | 0.93 |
| ST2 | 65.15 | 21.76 | 0.95 |
| VITS-R | 68.47 | 19.70 | 0.86 |
| VITS | 68.99 | 19.67 | 0.86 |
| ANC | 73.62 | 19.56 | 0.79 |
| XTTS | **73.65** | 18.52 | 0.86 |
| REF | 76.39 | 18.05 | 0.81 |

### A.5 VISUALIZING MUSHRA DISTRIBUTIONS

In Section 4, we discussed the distribution of MUSHRA scores across raters for Hindi using Figure 1. Similarly, in Figure 10 , we visualize the MUSHRA scores per rater across the three systems for Tamil. The Figure 11 and Figure 12, visualizes the MUSHRA scores for each utterance, averaged across raters, for Hindi and Tamil respectively.

### A.6 LIMITATIONS

Our study focuses on human evaluations for Hindi and Tamil, representing a major Indo-Aryan and Dravidian language, respectively. However, we did not extend our analysis to English, a widely spoken and diverse language. This limitation is due to the scope of our current research and resource constraints. Future studies should include evaluations in English to generalize our findings across different language families and understand how language-specific characteristics might influence TTS evaluation outcomes. This broader analysis could provide more comprehensive insights into the applicability and robustness of our proposed MUSHRA variants across diverse linguistic contexts.

| Criteria | To Mark |
|---|---|
| Mild Pronunciation | * Mark number of mild pronunciation errors.
* If no errors, mark 0 here.

A mild pronunciation error is where any character, for example an "r" or "t", is half-pronounced and not fully clear. |
| Severe Pronunciation | * Mark number of severe pronunciation errors. If no errors, mark 0 here.

A severe pronunciation error is where any character such as "r" or "t" is skipped/ mis-pronounced. |
| Unnatural Pauses, speedup or slowdown | * Mark number of places where there was unnatural pauses/speedup/slowdown in audio. |
| Liveliness | * Mark 100 if human-like
* Mark 85 if semi-expressive/ semi-enthusiastic/ semi-lively
* Mark 70 if robotic/monotonic

You may adjust scores in-between based on opinion. |
| Voice Quality/Clarity | * Mark 100 if perfect human like voice quality
* Mark 85 if slight digitalness in voice
* Mark 60-70 if high digitalness/persistent robotic voice

You may adjust scores in-between based on opinion |
| Rhythm | * Mark 100 if human-like
* Mark 85 if slightly fast/slow
* Mark 60 if too fast/slow

You may adjust scores in-between based on opinion |
| Digital Artifacts | * Mark number of digital artifacts heard in audio. If no artifacts, mark 0 here.

A digital artifact could be a "click" sound, "pop" sound, digital vibration in pauses, etc. |
| Sudden Energy Fluctuations | * Mark number of regions in which the energy, rhythm, pitch of the speech suddenly or irregularly change.

* Mark 0 here if no such changes noticed. |
| Word Skips | * Mark the number of words the model has skipped. If no skips, mark 0 here. |

Figure 9: Guidelines presented to raters across multiple evaluation criteria in the MUSHRA-DG Test.

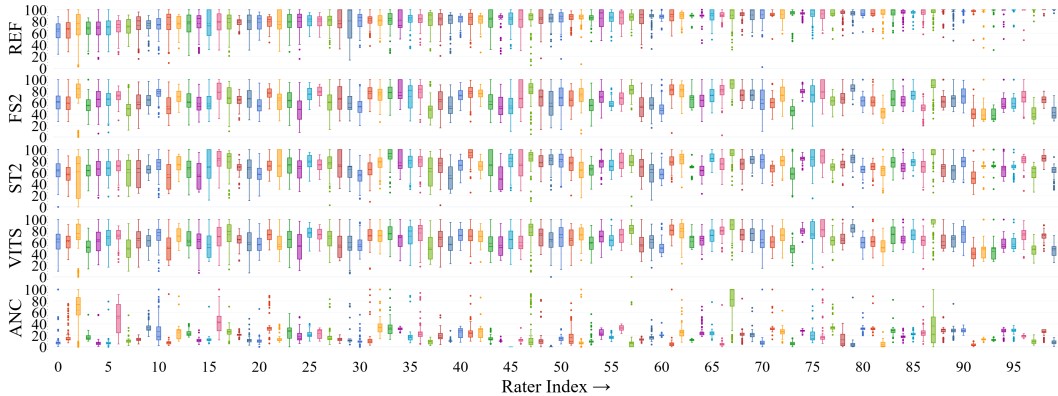

Figure 10: Visualization of the MUSHRA score distributions per rater across three systems— FS2, ST2, and VITS, along with the reference (REF) and anchor (ANC) for Tamil. Each boxplot represents ratings (0-100) across all test utterances for a system by one rater. The substantial heights of some boxplots indicate significant variance in the scores of that rater. The variation in boxplot means across raters suggests a high level of inter-rater variance

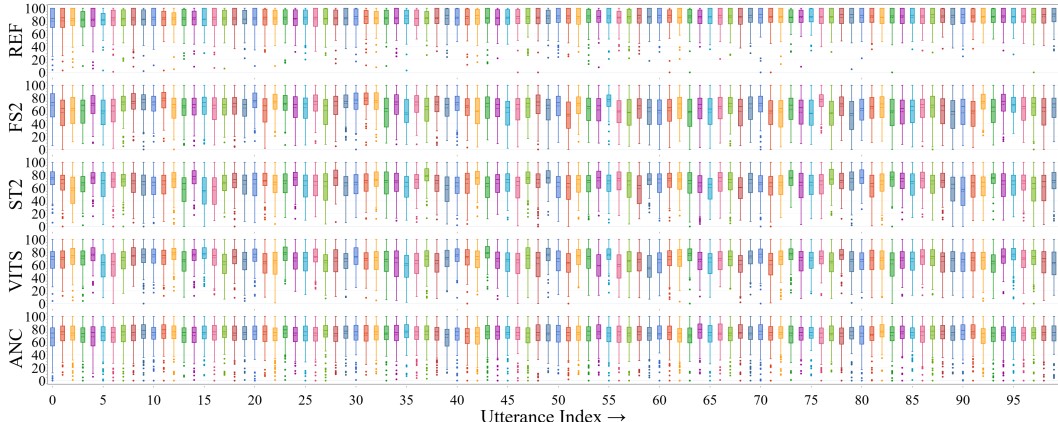

Figure 11: Visualization of the MUSHRA score distributions per utterance across three systems— FS2, ST2, and VITS, along with the reference (REF) and anchor (ANC) for Hindi. The X-axis represents each of the 100 utterances. Each boxplot represents ratings (0-100) across all raters for a system for a given utterance. The substantial heights of some boxplots indicate significant variance in the scores given by different raters for a single utterance.

## A.7 ETHICAL CONSIDERATIONS

We prioritized ethical conduct throughout our research. The 471 human listeners involved in the study provided informed consent before participating in the evaluation, recruited through professional data annotation agencies. These agencies verified participant language proficiency for task relevance. We established an education criterion of completing grade 12 (Indian system) to ensure participants' ability to accurately annotate audio content. Participants were compensated fairly for their time and expertise, following industry standards. They were also fully informed about the study nature, procedures, and their right to withdraw at any point without consequence.

We strived for inclusivity and bias mitigation. Participants came from diverse demographic backgrounds, and for Hindi and Tamil evaluations, we recruited only native speakers to capture the subtle linguistic and cultural nuances of each language. To minimize rater burden and bias in the new MUSHRA test variations, we prioritized user-friendliness and transparency in the design, providing clear guidelines.

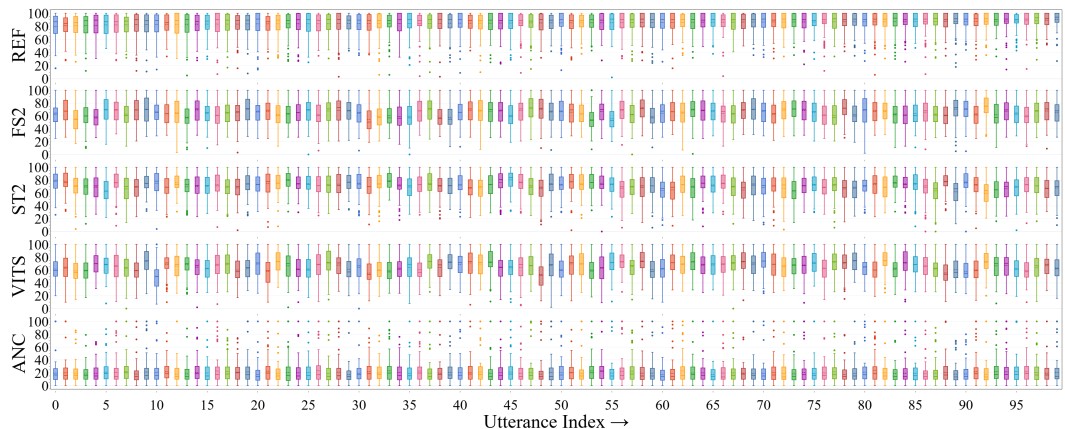

Figure 12: Visualization of the MUSHRA score distributions per utterance across three systems—FS2, ST2, and VITS, along with the reference (REF) and anchor (ANC) for Tamil. The X-axis represents each of the 100 utterances. Each boxplot represents ratings (0-100) across all raters for a system for a given utterance. The substantial heights of some boxplots indicate significant variance in the scores given by different raters for a single utterance.

We release the evaluations dataset, which includes 47,100 human ratings, under CC-BY-4.0 license after careful consideration of privacy and ethical use. Identifiable information about the participants was anonymized to protect their privacy. We encourage the use of this dataset for advancing TTS evaluation metrics, emphasizing that it should be used responsibly and ethically, adhering to principles of transparency and fairness. Finally, we acknowledge that our study focuses on Hindi and Tamil, and we recognize the importance of extending such evaluations to other languages, including English, to generalize our findings. Future research should continue to explore these ethical dimensions, ensuring that the development and evaluation of TTS systems are conducted with respect for the diversity and rights of all participants involved.

