# OpenReview forum: "Rethinking MUSHRA: Addressing Modern Challenges in Text-to-Speech Evaluation"
_ICLR.cc/2025/Conference — ICLR 2025 Conference Withdrawn Submission_

### Official Review · Reviewer_hbtx · 2024-10-31

**Soundness:** 3
**Presentation:** 3
**Contribution:** 3
**Rating:** 6
**Confidence:** 3

**Summary:**

his paper identifies two significant limitations of MUSHRA evaluation for TTS systems:

1. **Reference-Match Bias**: The use of explicit reference samples can lead raters to assign lower scores to synthesized samples that deviate from the reference, even if the quality is high.
2. **Judgment Ambiguity**: MUSHRA tests lack fine-grained guidance, making it difficult for raters to provide consistent evaluations.

To address these shortcomings, the authors propose two solutions:

1. Not explicitly identifying the human reference to the rater.
2. Clearly listing 9 key aspects of TTS evaluation (such as pronunciation errors and unnatural pauses) for raters to assess, followed by calculating the final MUSHRA score using a predefined formula.

Experimental results from three TTS systems in two Indian languages (Indic and Tamil) demonstrate that these proposed solutions effectively alleviate the issues associated with MUSHRA. The results show more consistent results with one-to-one comparisons between TTS systems and reference samples assessed by CMOS, and a reduction in score vari among different raters. Additionally, the complete dataset, comprising 47,100 ratings from 471 listeners evaluating three TTS systems, is publicly available for future research in this area.

**Strengths:**

- This paper highlights a significant issue with the foundational assumption of MUSHRA scores: it presupposes that a real reference sample should consistently receive high scores. However, this may not always be the case, as modern TTS systems can sometimes outperform human references.
- The authors conduct a thorough analysis of MUSHRA test results, providing both qualitative and quantitative evidence to demonstrate the two major shortcomings of the original MUSHRA framework.
- In addition to addressing the two primary shortcomings, the paper systematically examines various aspects of MUSHRA evaluation, including:
  - A comparison of MUSHRA and CMOS results
  - The sensitivity of MUSHRA scores to the number of raters and utterances
  - Protocols for rejecting raters
  - Different strategies for establishing anchors
- The proposed MUSHRA variants are straightforward and effectively address the identified shortcomings, enhancing the reliability of the evaluation process.

**Weaknesses:**

- While the issues with MUSHRA are language-agnostic, this paper focuses exclusively on TTS systems for Indian languages, which are trained on relatively small datasets. The findings would be more compelling if the authors included results from widely-used datasets of high-resource languages and publicly available pre-trained English (or any other high-resource language) TTS systems with verified quality.
- The nine aspects selected for MUSHRA-DG appear to be arbitrary, as the authors do not provide justification or references for their choices, leaving unanswered why some other common aspects (such as word repetition) were not included.
- As already noted in Section 6.1 (line 410), even though the reference-matching bias issue is mitigated, discrepancies remain between the results of CMOS, MUSHRA-NMR, and MUSHRA-DG-NMR, particularly for the best-performing ST2 system in Tamil. More detailed explanations regarding these differences will be appreciated.

**Questions:**

In line 256, the authors highlight an issue with MUSHRA tests, stating that “the individual box-plots have a high variance indicating that the same rater rates the system very differently across utterances.” However, this variance could also result from failures in the generation process, particularly since the variance in the reference (REF) is smaller than that of any TTS system. Could the authors clarify why they interpret this observation as an issue with rater consistency rather than a reflection of generation quality?

---

### Official Review · Reviewer_S4bg · 2024-10-31

**Soundness:** 2
**Presentation:** 2
**Contribution:** 2
**Rating:** 3
**Confidence:** 4

**Summary:**

The paper analyzed the MUSHRA test for TTS evaluations. The authors proposed a dataset and two refined methods based on MUSHRA.

**Strengths:**

The paper analyzed the advantages and disadvantages of MUSHRA test in detail.

**Weaknesses:**

The paper used nearly all space to rethink the MUSHRA test. However, the proposed methods seem to be too limited and hard to reproduce.

**Questions:**

1. Is the proposed method robust to other languages like English and Spanish?
2. In the MUSHRA-DG test, there are 9 criterias listed in the figure 9. Are they easy to be evaluated? Can the evaluation task assigned to general people like the MOS test?

---

### Official Review · Reviewer_nzGH · 2024-11-03

**Soundness:** 2
**Presentation:** 2
**Contribution:** 2
**Rating:** 3
**Confidence:** 5

**Summary:**

The authors want to assess the reliability of MUSHRA which is one of  the most popular subjective evaluation techniques in TTS. They do so by conducting extensive evaluation in two languages, Tamil and Hindi. They identify two shortcomings and they propose two variants of MUSHRA that address these shortcomings. They also provide MANGO, a dataset of 47,100 ratings from 471 listeners across Hindi and Tamil.

**Strengths:**

It is important to discuss about the evaluations of TTS systems. It is an open-topic and given the subjective nature of these evaluations, it is useful to understand the shortcomings of the used methodology and hopefully agree as a community to improved guidelines. Some of the claims make in the paper make sense, like the need of clear guidelines to help with ambiguity, the importance of a sufficient number of listeners or that a hidden reference is a better choice.

I also think that in the TTS systems compared, there is a good choice of TTS systems with a variety in modeling approaches and state-of-the-art results.

**Weaknesses:**

The biggest drawbacks of the paper for me is the lack of technical details on how the compared TTS systems are trained and on the data used. Concerning the systems, I understand that an open-source version of each one is used, but did you do any changes on top of them? Did you use them as is and did inference? Did you do fine-tuning on your data?

Similarly there is a lack of details on the data used. If you did some kind of fine-tuning, what data did you use? What about inference? Was it done on the MANGO dataset? What style/genre of data is there included in your training and test sets? How many data did you use? How complex are your data?

Without knowing these technical details, it is very difficult to draw any conclusions on the validity of the claims that are made. This is particularly important because it is a theoretical study, thus the findings must give important and new insights for it to have a value. Plus the suggestions must be useful for a large audience.

My other concern is that even the conclusions that are drawn are of limited novelty and I don't see how the suggestions of the authors add a value to the MUSHRA evaluation that can radically change how it is done from now on.

Some further more detailed comments:
-“Given that state-of-the-art TTS systems are able to reach quality on par with references, one would expect a much smaller gap between the reference and systems. “ -> That is too big of an assumption, not always the case. Again it heavily depends on the exact version of a system and on the used data. The fact that this claim is made in one paper (StyleTTS2) and for one specific corpus (LJSpeech ), does not make it a general claim for state-of-the-art TTS systems, not even for the same system (StyletTS2) on other corpora.
-Table 2: Pretty confusing to follow when first introduced. the line "ANCH" correspond to a different type of ANCH per language which is not explained until section 4.5, while Table 2 is first introduced in 4.1 I believe.
-“CMOS indicates that the outputs synthesised by VITS and ST2 are very close in quality to the reference in Hindi and Tamil respectively “  -> Not that close for Tamil, is it? Much closer for Hindi. Is this expected?
-“Clear guidelines can help with ambiguity”-> sure, but isn't it obvious? What is new in this observation?
-Section 4.3: Somehow interesting but obvious results that everyone that has worked with MUSHRA already knows empirically. The number of listeners and utterances in most practical cases is a question of cost.
-It is recommended not to use anchor, but anchor is not a commonly used practice in TTS evaluations in any case AFAIK.
-I agree on the merit of detailed guidelines. But it has to be discussed if training of the listeners is required and if this increases the MUSHRA cost.

**Questions:**

Please see the Weaknesses section with detailed observations, suggestions and questions on each comment for the authors to respond to.

---

### Official Review · Reviewer_wTSx · 2024-11-04

**Soundness:** 2
**Presentation:** 3
**Contribution:** 3
**Rating:** 5
**Confidence:** 3

**Summary:**

This paper discusses two shortcomings of MUSHRA: (1) reference-matching bias and (2) judgment ambiguity. The authors first created a large-scale evaluation dataset called MANGO in Hindi and Tamil. Based on this dataset, they analyzed MUSHRA scores from various perspectives. They then propose two variants to tackle these problems. In the first variant, NMR (No Mentioned Reference), subjects are less likely to strictly focus on aligning their scores with the reference. In the second variant, DG (Detailed Guidelines), subjects are provided with fine-grained evaluation criteria to score each audio sample, thus reducing score variance. Both variants are shown to effectively mitigate these issues.

**Strengths:**

This paper addresses two important issues in subjective evaluation, which is an essential part of developing speech synthesis systems. The authors have created a large-scale dataset called MANGO, which I expect could be utilized in several applications in speech quality assessment beyond the analysis presented in this paper. They study existing MUSHRA scores from various perspectives, including reliability, sensitivity, and rejection mechanisms. The proposed approaches effectively address these issues according to the experimental results.

**Weaknesses:**

The MUSHRA-DG variant would significantly increase the time cost and difficulty of scoring for human subjects. Allowing subjects to adjust scores if they feel the final MUSHRA score does not match their expectations seems questionable to me, as it may encourage them to revise sub-scores only to fit the final outcome instead of focusing on the fine-grained scores they should judge fairly. Additionally, the combination of the two proposed approaches does not appear to be as effective as the individual ones.

**Questions:**

1. In Section 6.2, the authors say that 14 participants were involved in conducting MUSHRA-DG. I wonder if this number of participants is sufficient, as in other experiments in the paper the authors included many more participants. Moreover, the experimental results also suggest that it is recommended to include 20 or more subjects in MUSHRA tests.

2. In Table 4, MUSHRA-DG-NMR seems to fail to distinguish between the evaluated TTS models, as they have very close mean scores (around 89). Is there any explanation for this phenomenon?

3. In the last paragraph of Section 5, the authors mention that the subjects can adjust the fine-grained scores if they find the final scores do not meet their expectations. This seems odd to me, as the main goal of designing fine-grained scores should be to relieve subjects from giving an overall score that contains too much ambiguity. Therefore, if I understand correctly, shouldn't we just ask subjects to score the fine-grained parts? Are there any results in the case where subjects are not shown the final scores?

---

### Note · Authors · 2024-11-29

**Comment:**

We sincerely thank all reviewers for their time and thoughtful feedback. While we sincerely appreciate the constructive insights provided by Reviewer wTSx and Reviewer hbtx, we were disheartened by certain comments from other reviewers.

Some of Reviewer nzGH's questions, such as, "What about inference? Was it done on the MANGO?" seem to reflect a lack of careful reading, as this is clearly addressed in the manuscript. Additionally, we were disappointed by the brief and discourteous review from Reviewer S4bg, who raised concerns about reproducibility without providing specific elaboration.

While we understand the emphasis on analyzing results in English, given its status as the world's most widely spoken language, we are unclear why evaluations in English are deemed essential for acceptance, especially as our work focuses on the fourth and seventeenth most spoken languages globally.

We take all criticism in good heart, viewing it as an opportunity to grow, and will strive to improve our manuscript, despite the significant expense and effort involved in subjective evaluations. We sincerely thank all reviewers once again for their time and comments and hope to return with a stronger submission in the future.

**Withdrawal Confirmation:**

I have read and agree with the venue's withdrawal policy on behalf of myself and my co-authors.